# Validation of a Short-Form Version of the Danish Need for Recovery Scale against the Full Scale

**DOI:** 10.3390/ijerph16132334

**Published:** 2019-07-02

**Authors:** Matthew L. Stevens, Patrick Crowley, Anne H. Garde, Ole S. Mortensen, Clas-Håkan Nygård, Andreas Holtermann

**Affiliations:** 1The National Research Centre for the Working Environment, 2100 Copenhagen, Denmark; 2Sydney School of Public Health, Faculty of Medicine and Health, University of Sydney, Sydney 2006, Australia; 3Department of Public Health, Copenhagen University, 1165 Copenhagen, Denmark; 4Section of Social Medicine, Department of Public Health, University of Copenhagen, 1123 Copenhagen, Denmark; 5Department of Occupational and Social Medicine, Copenhagen University Hospital, 4300 Holbæk, Denmark; 6Unit of Health Sciences, Faculty of Social Sciences, Tampere University, 33100 Tampere, Finland; 7Department of Sports Science and Clinical Biomechanics, University of Southern Denmark, 5230 Odense, Denmark

**Keywords:** intermediate outcomes, sustainable employment, occupational health, work ability, aging, short-form validation, need for recovery, criterion validity, construct validity, content validity, responsiveness

## Abstract

Introduction: The Need for Recovery (NFR) Scale facilitates the understanding of the factors that can lead to sustainable working and employability. Short-form scales can reduce the burden on researchers and respondents. Our aim was to create and validate a short-form Danish version of the NFR Scale. Methods: Two datasets were used to conduct the exploratory and confirmatory analyses. This was done using qualitative and quantitative methods. The exploratory phase identified several short-form versions of the Danish NFR Scale and evaluated the quality of each through the assessment of content, construct and criterion validity, and responsiveness. These evaluations were then verified through the confirmatory analysis, using the second dataset. Results: A short-form NFR scale consisting of three items (exhausted at the end of a work day, hard to find interest in other people after a work day, it takes over an hour to fully recover from a work day) showed excellent validity and responsiveness compared to the nine-item scale. Furthermore, a short-form consisting of just two items also showed excellent validity and good responsiveness. Conclusion: A short-form NFR scale, consisting of three items from the Danish NFR Scale, seems to be an appropriate substitute for the full nine-item scale.

## 1. Introduction

Sustainable working ability and employability are important challenges facing modern economies. The current combination of an aging population, increasing socioeconomic health inequalities, and significant proportions of the population with limited ability to work, brings into question the sustainable working years of the general population [1]. To achieve this, a proper understanding of the factors that lead to sustainable employability is required, and this, in turn, requires precise and accurate outcome measures [2].

Outcome measures, like sickness absence and disability pensioning, signify fundamental constructs in the evaluation of sustainable employability. However, they also represent hard-end outcomes of a negative trajectory for the individual that could be delayed or prevented with effective early interventions. In order to appropriately target these interventions, intermediate outcomes—that identify individuals with an increased risk of the aforementioned hard end outcomes—are needed. The intermediate outcome measures also make research considerably more cost-efficient, since they require shorter follow-up times and often fewer participants [3].

Another important aspect of intermediate outcome measures, apart from their precision and relationship to important health outcomes, is their burdensomeness. This can represent the burden that collecting data places on the researchers, but more typically, represents the burden placed upon participants. For instance, the burdensomeness of participant-reported outcome measures (e.g., questionnaires) is usually defined by the number and complexity of the questions asked. This burdensomeness is particularly important for longitudinal cohort or intervention studies, where participants are asked to complete an extensive array of questionnaires at multiple time points. Reducing burdensomeness (e.g., through implementing shorter questionnaires) has been shown to significantly increase participant response rates [4,5,6] and, as such, the shorter and more accessible a questionnaire can be whilst still maintaining its validity, the better [7].

A commonly utilized intermediate outcome measure in the work-health-sustainability field is the Need for Recovery (NFR) Scale [8]. The NFR Scale was first developed in the Netherlands in 1994 as part of the Dutch Questionnaire on the Experience and Evaluation of Work [9] and was designed as a short-term outcome capable of predicting long-term work-related fatigue symptoms (e.g., burnout) [10]. It has strong content validity, being closely related with other subjective measures of fatigue, such as the Checklist for Individual Strength (CIS-20; *r* = 0.66 to 0.77) and emotional exhaustion (*r* = 0.84), good internal (rho = 0.86–0.87; α = 0.88) and test-retest (intraclass correlation coefficient (ICC) = 0.68–0.80), reliability and is sensitive to detecting change [8,10]. Most importantly, the NFR Scale also functions as a good predictor of sickness absence—being an important risk-factor for work absenteeism in multiple workgroups [11,12]. Since its original development in the Netherlands, it has gained popularity amongst the work-health community globally and has been translated into several other languages, including Portuguese [13,14], Italian [15], Taiwanese [16], and Danish [17,18].

To facilitate the use of the Danish NFR Scale in research, it would be useful to develop a short-form version that reduces the number of items to as few as possible, whilst also maintaining the scale’s validity. Such a reduction has been successfully conducted with the Work Ability Index (WAI), which was recently reduced down to a single item version [19,20]. However, this short-form validation has not yet been performed for NFR. Doing so would greatly decrease the burden of the NFR Scale in research and thus increase its feasibility of use in future cohorts and studies. Therefore, the aim of this study is to create and validate a shortened version of the Danish NFR Scale that is an adequate representation of the full scale. A secondary aim is to validate a specific shortened Danish version of the NFR Scale that has been used in previous studies [21,22] as an adequate representation of the full scale.

## 2. Materials and Methods

This validation study was conducted using data from two previous workplace interventions—the Participatory Intervention on Physical and Psychosocial resources of Industrial workers (PIPPI) and the Prioritized Working Hours ([Prioritet Arbejdstid]; PRIO) project. PIPPI was a cluster-randomized controlled trial that investigated the effectiveness of a participatory physical and psychosocial intervention on NFR among industrial workers. This trial was pre-registered with the Danish Data Protection Agency register (2013-54-0329) and in the International Standard RCT Register (ISRCTN76842602). Ethical approval was provided by the Ethical Committee for the regional capital in Denmark (H-2-2013-FSP13). PRIO was a non-randomized controlled trial that investigated the effectiveness of self-rostering on NFR among workers in the healthcare sector. Approval for this trial was provided by the Danish Data Protection Agency (2008-54-0458). Full details for both these trials have been previously published [17,18,23].

### 2.1. Participants

Participants in PIPPI were recruited from three large Danish workplaces in the manufacturing and production sectors. Inclusion criteria for workplaces required a minimum of 100 employees involved in primarily manual labor, who work within a team based structure involving a cooperative relationship between different organizational levels. Moreover the workplaces must have been willing to implement the intervention activities of the PIPPI trial, and must have reflected the geographical and organizational distribution of Danish production companies. The inclusion criteria for eligible workers was to work ≥20 h/week and to provide informed consent to participate in the study.

Participants in PRIO were recruited from nine Danish companies covering 28 different workplaces, all involving shift work. These workplaces were contacted through public advertising, meetings, and personal contacts. Inclusion criteria required that the workplace was in the planning stage of implementing self-rostering (*n* = 14). Workers who chose to respond to the questionnaire were included in the PRIO study. All participants from PIPPI and PRIO that provided any information regarding NFR at baseline were included in this study.

### 2.2. Outcomes and Data Collection

Collected outcome measures used in this study include the Danish NFR Scale, several related measures for the assessment of construct validity, and basic demographic information. The Danish NFR Scale is a 9-item Likert scale with five response categories: “Never”; “Rarely”; “Some of the Time”; “Most of the Time” and “Always”. The individual sum of these scores is converted to an index from 1 to 100, where 100 indicates the maximum requirement for recovery. All items included in the Danish NFR Scale and their English translations are provided in Table 1. A comparison of these items with the original questionnaire is provided in the online Appendix A. The Danish NFR Scale was collected at baseline and 12 months in both PIPPI and PRIO projects. The measures collected for the assessment of construct validity were collected at baseline in the PIPPI dataset and included general health and wellbeing, the number of days with limitations due to pain, work-ability and perceived exertion at work [24,25,26,27]. A full list of these items is contained in Table 2.

### 2.3. Validation Process and Statistical Analyses

The validation process and psychometric definitions (e.g., content, construct, and criterion validity) followed the guidelines outlined by Goetz et al. [28] and the COSMIN group [2]. Accordingly, this validation contained two phases. The initial phase was an exploratory analysis, using the PIPPI dataset aimed at identifying short-form versions of the Danish NFR Scale that adequately maintained the scale’s psychometric qualities. In this phase, we first examined each item individually to determine its suitability for inclusion in a short-form NFR scale. From these results we then developed short-form scales and tested them against the full scale, in order to determine how well these short form versions represented the full scale. The second phase was a confirmatory analysis using the PRIO dataset, aimed at validating the short-form NFR scale/s identified in the exploratory analyses. This process is illustrated in Figure 1. To fulfil the primary aim, all 9-items of the Danish NFR Scale were considered for inclusion in a short-form version. However, to fulfil the secondary aim, only items 1, 2, and 9 were considered. All statistical analyses were conducted using R v3.4.3 (R Foundation for Statistical Computing, Vienna, Austria) [29]. The R-packages that were used for analysis were the ‘GPArotation’ [30], ‘dplyr’ [31], ‘ggplot2’ [32], and ‘psych’ [33] packages.

Although not conducted prior to the analyses (due to the data being sourced from previously conducted studies), a post-hoc calculation of sample size was carried out using our analysis of criterion validity (the primary measure of whether the short-form scales adequately represent the full scale). In this case, the minimum sample size required to achieve 80% power, given an agreement (ICC) of 0.75 and an alpha of 0.05, is 75 [34]. With the number of participants included in our analyses (1109 counting both exploratory and confirmatory analyses) we believe our sample size is more than adequate for the analyses conducted.

### 2.4. Exploratory Phase

The exploratory phase began with an assessment of missing cases and response distributions across categories of the NFR questionnaire. Content (face) validity was then assessed by the author group for both the scale as a whole and for each individual item. This involved a subjective assessment of the meaning of each item in terms of how it relates to the concept of NFR. This assessment by the author group was complemented by unstructured interviews conducted with the investigators of each study (PIPPI and PRIO). These interviews were used to gain an understanding of participants’ views on the interpretation and accessibility of each item. These interviews began openly, asking what participants thought of the items in general; but then later focused on whether participants found any of the items either, not relevant, or difficult to understand, and which items fell under these categories and why. The interviews were then followed by an exploratory factor analysis (EFA). For the factor analysis, the component eigenvalue threshold was set at 1 and, as such, only components above this value were considered significant. To optimize the reduction of items, discrimination parameters and item difficulty for each item were also calculated based on item response theory [2].

Following the interviews and EFA, several short-form versions of the Danish NFR Scale were developed and compared to the 9-item scale with regard to their construct validity, criterion validity, and responsiveness. Development of these short form versions was based upon the previous analysis, and involved subjective decisions by the author group about which item(s) would be most likely to accurately represent the concept of NFR generally and the full NFR scale specifically. The analysis of construct validity between the 9-item scale and the developed short-form versions was conducted by comparing their correlations to eight related constructs (Table 2). Correlations were calculated using either Pearson’s r or Kendall’s tau, depending upon the data distribution. Confidence intervals for Kendall’s tau were obtained through bootstrapping. To assess the criterion validity and responsiveness of the short-form versions, Bland-Altman plots were developed and inter-class correlation (ICC) scores calculated. For the ICC, values between 0.40 and 0.59 were considered fair, values between 0.60 and 0.74 were considered good and values above 0.75 were considered excellent [26]. For the purposes of validation, it was decided apriori that the construct validity of the short-form NFR versions and the full Danish NFR Scale to the related measures should not differ by more than 0.1 (i.e., Δ rho/tau ≤ 0.1) and that the criterion validity and responsiveness between the scales should be excellent (i.e., ICC ≥ 0.75).

### 2.5. Confirmatory Phase

To confirm the findings of the exploratory phase in the PIPPI dataset, the confirmatory phase replicated the analyses conducted for criterion validity and responsiveness conducted in the exploratory phase in the separate PRIO dataset. This was conducted for three short-form NFR scales identified by the author group as being most suitable, based upon the findings of the exploratory analysis.

## 3. Results

Of the 415 participants in the PIPPI intervention, 344 provided data used in this analysis (242 male, age = 44 ± 10.4 years, body mass index (BMI) = 26.6 ± 4.4). Of the 811 participants that participated at baseline in PRIO, 765 provided data for this study (74 male, 43 ± 10.8 years, 25 ± 3.9 BMI). Full demographic details are presented in Table 3. Further detail on PIPPI participants regarding their response distributions for the items used to assess construct validity is provided in Table 4.

### 3.1. Exploratory Analyses

Cases with missing responses to any NFR items were handled through list-wise deletion. The distribution of missing responses is presented in Table 5. The response distributions and covariance matrix for each item of the 9 item scale are provided in the Appendix A.

#### 3.1.1. Content Validity and Item-Accessibility

The assessment of content validity of the items of the Danish NFR Scale suggested two primary factors:
Factor 1: Recovery of mental resources—items 1, 4, 5, 6, and 8. These items refer to constructs, such as ‘trouble relaxing’, ‘trouble concentrating’, ‘hard to show interest in others’, and ‘a need for being left alone after work’. These phrases can predominantly be linked to increased mental stress and fatigue, apathy, and irritability; all being symptoms of drained mental resources.Factor 2: Recovery of physical resources—items 2 and 9. These items use words, such as ‘exhausted’ or indicate ‘tiredness’ too great to initiate other activities. Both of these items could be interpreted as referring to the depletion of physical resources.

Two items (items 3 and 7) did not fit into these categories. Item 3 refers to ‘feeling fresh’, which could be interpreted as either mentally or physically ‘fresh’ or perhaps a combination of both. Similarly, item 7 refers simply to ‘recovery,’ which could be either mental or physical recovery.

The unstructured interviews identified that item interpretability was poor for items 3 and 4; reporting that many participants either, did not understand these items or felt they were not relevant. For example, shift workers often finish work at odd hours (e.g., morning) and therefore feeling ‘fresh after dinner’ (Item 3) did not relate to how fatigued they felt after work. Similarly, those not working typical hours also reported confusion as to how to interpret ‘only had one day without work’ (Item 4). Accordingly, these questions were also those which had the most missing responses, as shown in Table 5. Moreover, many of the participants reported never being able to ‘really relax’, due to domestic responsibilities (e.g., single parents looking after children) or from anxiety, due to previous trauma (e.g., refugees).

#### 3.1.2. Factor Analysis

The EFA identified two primary factors (Eigen values = 3.97, 0.95). This finding matches the assessment of face validity. Items 1, 5, 6, and 8 loaded primarily on factor 1 (recovery of mental resources; *β* = 0.50, 0.54, 0.83, 0.73, respectively). Items 2, 3, and 9 loaded primarily on factor 2 (recovery of physical resources; *β* = 0.76, −0.46, and 0.47, respectively). Item 7 loaded on both factors 1 and 2 (*β* = 0.43 and 0.41 respectively). While item 4 did not clearly load on any factor. Full details for factor loading are provided in the Appendix A.

Response curves developed using item response theory showed that item 7 had the highest discriminative validity (Figure 2). Items 2, 5, 6, 8, and 9 also showed reasonable discriminative validity. Item 4 showed the poorest discriminative validity, followed by item 3. Item information scores for each of the 9 items are provided in the Appendix A.

#### 3.1.3. Development of the Short-Form Versions

From the assessments of face validity, item accessibility and factor analyses of the Danish NFR Scale, 5 short-form versions of the Danish NFR were developed.

Items 1, 2 and 5–9: This short-form version was developed by dropping only those items (Items 3 and 4) that were deemed undesirable. This judgement was based upon the results from the unstructured interviews, which revealed that many participants did not consider them meaningful and/or found these two items difficult to interpret.

Items 2, 6, and 7: The next stage focused on minimizing the number of items, retaining those items with the highest discriminative ability, whilst also taking into account the need to maintain a balance between items that loaded on the two identified factors (physical/mental recovery) and the scale’s overall face-validity. These 3 items were chosen as they were the 3 items with the highest discriminative ability, which maintained a balance of factor loadings identified in the assessment of content validity.

Items 2 and 6: A two-item version, that included items 2 and 6, was then developed utilizing the same rationale as above. Although item 7 had the highest discriminative ability, it was decided that maintaining the balance between factors identified in the assessment of content validity was more important.

Item 7: To assess the validity of a single item version of the NFR scale. This item showed the highest discriminative ability on the basis of response curve analyses (Figure 2) and factor balance, as highlighted during our factor analysis.

Items 1, 2 and 9: This short-form version was created for completion of the secondary aim—to assess the validity of a short-form used in previous trials [21,22].

Items 2 and 9: To further minimize the number of items, this version dropped item 1, due to its poor discriminative ability compared to items 2 and 9, as represented by the response curves in Figure 2.

#### 3.1.4. Construct Validity

The analyses for construct validity showed small to moderate correlations [35] between the various NFR scales and the related constructs (pain, work ability, perceived health etc.; Table 2). However, the difference between the NFR scales and any specific related measure was never greater than 0.1—suggesting the idea that all of the short-form NFR scale versions developed, in fact, measure the same construct. The strongest correlation was between the NFR scales and ‘feeling rested on waking’ (r = −0.48 to −0.39), whilst the weakest correlation was with ‘perceived exertion’ (r = 0.17 to 0.26). Complete results are presented in Table 6.

#### 3.1.5. Criterion Validity and Responsiveness

Correlations for criterion validity and responsiveness of the short-form versions to the 9-item scale ranged from 0.66 to 0.92 and 0.67 to 0.94 respectively. Two short-form versions of the Danish NFR Scale met our criteria (ICC ≥ 0.75) for both criterion validity and responsiveness (Items 2, 6, and 7; Items 1, 2 and 9). Of the remaining versions developed, two (Item 7; Items 2 and 9) did not meet our criteria for criterion validity and three (Items 2 and 6; Item 7; Items 2 and 9) did not meet our established criteria for responsiveness. Complete ICC scores and confidence intervals are presented in Table 7. Bland Altman plots are provided in the online Appendix A.

### 3.2. Confirmatory Analyses

Three short-form versions (Items 2, 6 and 7, Items 2 and 6, Items 1, 2, and 9) were carried forward for confirmatory analysis in the PRIO dataset. In the PRIO dataset, all three short-form versions met our pre-specified criteria for criterion validity with ICCs of 0.88, 0.82 and 0.86, respectively. However, only one short-form version (Items 2, 6 and 7) met our criteria for responsiveness with an ICC of 0.80. The two other short-for versions tested (Items 2 and 6, Items 1, 2, and 9) had ICCs of 0.72 and 0.73 respectively. These scores and confidence intervals are presented in Table 8. Bland Altman plots are provided in the Appendix A.

## 4. Discussion

### 4.1. Summary of Findings

Of the developed short form versions only one (consisting of items 2, 6, and 7; At the end of my work day I am exhausted, I find it hard to show interest in other people when I have just come home from work, It takes me over an hour before I am fully recovered after a work day, respectively) met all pre-specified criteria for construct and criterion validity and responsiveness. Additionally, two short form versions (consisting of items 2 and 6, and of items 1, 2 and 9; Table 1) met our pre-specified criteria for construct and criterion validity but achieved only good responsiveness.

### 4.2. Strengths and Limitations of the Study

A major strength of this study is the comprehensive validation process, which adhered to the established guidelines [28] and recommendations of the COSMIN group [2], and included both qualitative and quantitative analyses conducted in a structured, transparent manner. Furthermore, the conduct of separate exploratory and confirmatory analyses in two diverse occupational groups helps to ensure the generalizability of results. However, this study also contains some clear limitations. One such limitation is our inability to interview respondents directly. Instead we relied on the recall of those involved in carrying out these previous trials. Despite this, we are confident that the views of participants on these items were adequately reflected as there was clear consensus from interviewees regarding which items lacked interpretability (i.e., items 3 and 4). A further limitation of our findings is that we did not assess the predictive capability of the short-form versions developed with regard to hard outcomes, such as sickness absence [12]; however, such analyses were outside the scope of this study.

### 4.3. Comparisons with Other Studies

To the best of our knowledge, this is the first study that has attempted to create and validate a short-form version of the NFR scale, Danish or otherwise. Although we succeeded in reducing the number of questionnaire items, we were unable to show support for reduction to just a single item, as has been the case for other scales—such as the Work Ability Index (WAI) [19,20]. Further shortening the Danish NFR Scale to include just a single item (item 7) resulted in an unacceptable reduction in criterion validity and responsiveness. To help explain why this reduction to a single item was not possible for the Danish NFR Scale, it may be useful to examine the reason why other scales, like the WAI, were successful.

The WAI was developed to answer the question, “How good workers are at present and in the near future and how able are they to do their job with respect to work demands, health, and mental resources?” [36]. Contained within the index is an overall item, for which, the respondent assigns a value between 0 and 10 to their current work ability relative to their lifetime best work ability [36]. It is this overall item that forms the single item measure of self-reported work ability often utilized today [17,18,36]. Unfortunately, the Danish NFR Scale does not contain an item that provides a similar overall assessment of ‘need for recovery’. Instead, the items of the NFR scale seem to measure more specific physical and/or psychological requirements for recovery which may explain why we were not able to identify a single item which adequately captured the general construct of NFR. Thus, if a single item Danish NFR Scale is required, it seems necessary to develop a new question that is able to capture the overall NFR construct.

### 4.4. Meaning and Implications of the Study

Valid short-form questionnaires reduce the burden on researchers and respondents alike, while simultaneously improving response rates [4,5,6]. Our primary aim was to create and validate such a short-form version for the Danish NFR Scale. In line with this aim, our analysis identified a short-form scale consisting of just three items from the original nine items; item 2—At the end of my work day I am exhausted, item 6—I find it hard to show interest in other people, when I have just come home from work, and item 7—It takes me over an hour before I am fully recovered after a work day. This version was the most statistically robust of the assessed versions, demonstrating excellent criterion validity and responsiveness. Moreover, our assessment of construct validity demonstrated that this short-form version is consistent with the full 9-item scale. Therefore, we assert that a short-form version consisting of items 2, 6, and 7, provides the best approximation of the underlying constructs captured by the full 9-item Danish NFR Scale.

A secondary aim of our study was to assess the validity of a specific short-form version of the Danish NFR Scale, used in previous trials [21,22]. This version consists of; item 1—I find it hard to relax after a working day, item 2—At the end of my work day I am exhausted, and item 9—After a workday I am too tired to begin other activities. Our findings show excellent criterion validity and good responsiveness for this short-form version, and since the construct validity was again consistent with the 9-item questionnaire, this short-form version is still likely to be an acceptable approximation of the underlying constructs captured by the full 9-item Danish NFR Scale.

### 4.5. Future Research

In order to further validate the short-form versions developed, future research should assess their predictive ability for hard-end outcomes, such as sickness absence and disability retirement. It may also be beneficial to develop a new question that is able to capture the NFR construct in a single item.

## 5. Conclusions

A short-form version of the Danish NFR Scale consisting of items 2, 6, and 7 (At the end of my work day I am exhausted; I find it hard to show interest in other people, when I have just come home from work, and It takes me over an hour before I am fully recovered after a work day, respectively) demonstrated excellent validity and responsiveness when compared to the Danish 9-item NFR Scale. We thus recommend this version be used where a short-form version is required. Any generalizations of these findings to other countries ought to be made with caution.

## Figures and Tables

**Figure 1 ijerph-16-02334-f001:**
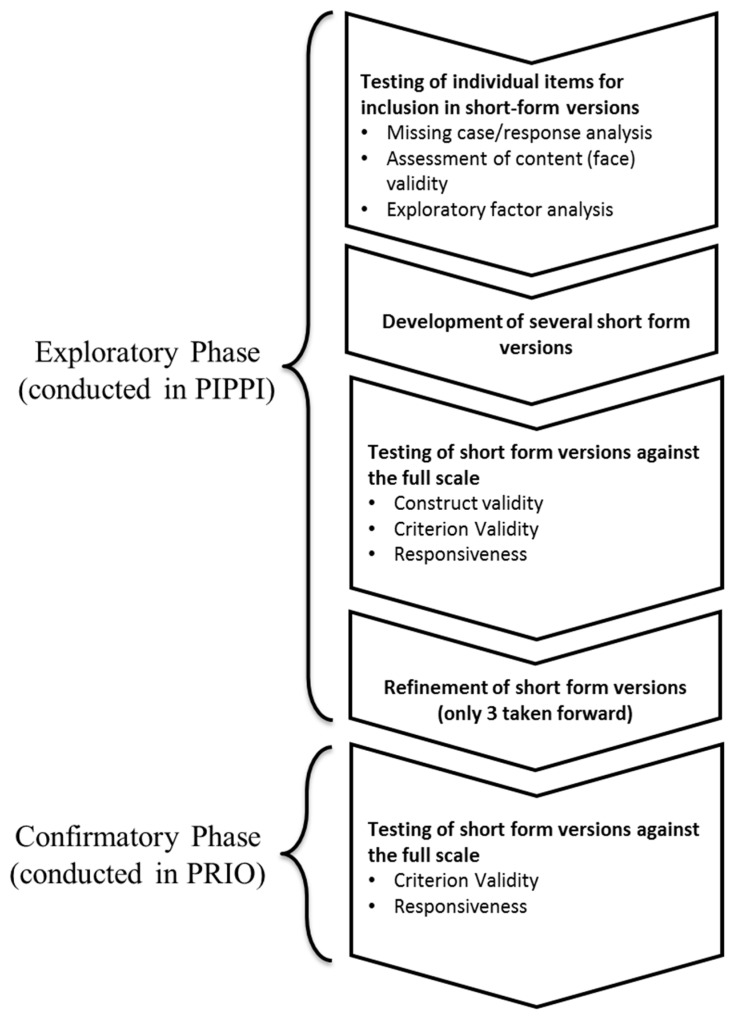
Flowchart showing the process for the development and testing of the short-form versions. PIPPI = Participatory Intervention on Physical and Psychosocial resources of Industrial workers; PRIO = Prioritized Working Hours [Prioritet Arbejdstid].

**Figure 2 ijerph-16-02334-f002:**
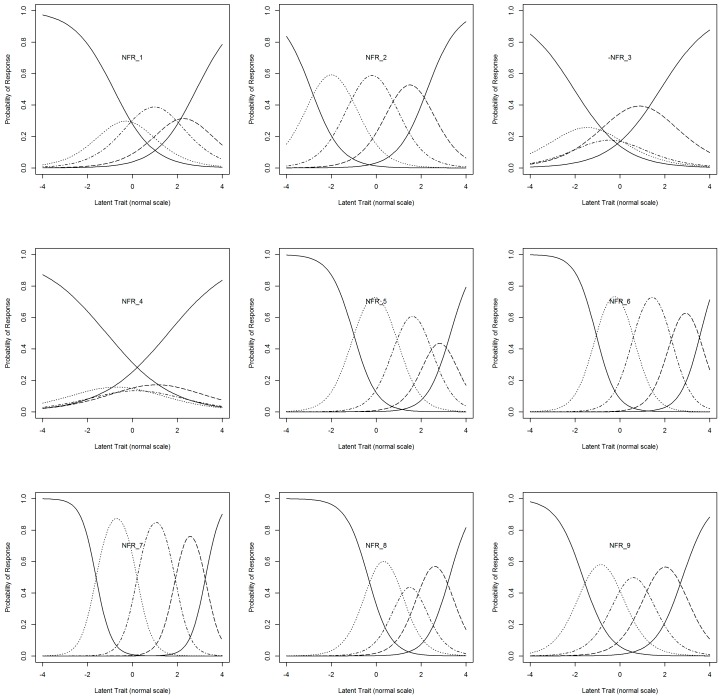
Response curves detailing the probability of identifying each level of the latent trait for a given item of the Danish Need for Recovery Scale. The latent trait represents the underlying constructs, upon which, the scale was built. For details of each item please refer to Table 1.

**Table 1 ijerph-16-02334-t001:** The nine items of the Danish Need for Recovery (NFR) Scale and their translation to English.

Item Number	Danish 9-Item NFR Scale	English Translation
Item 1	Jeg har svært ved at slappe af efter en arbejdsdag	I find it hard to relax after a working day
Item 2	I slutningen af min arbejdsdag er jeg udmattet	At the end of my work day I am exhausted
Item 3	Jeg føler mig frisk efter aftensmad	I feel fresh after dinner
Item 4	Jeg slapper ikke ordentlig af, hvis jeg kun har en dag uden arbejde	I do not normally relax, if I have only had one day without work
Item 5	Jeg har problemer med at koncentrere mig i timerne efter, at jeg er kommet hjem fra arbejde	I have trouble concentrating in the hours after I come home from work
Item 6	Jeg har svært ved at udvise interesse for andre mennesker, lige når jeg er kommet hjem fra arbejde	I find it hard to show interest in other people, when I have just come home from work
Item 7	Det tager mig over en time, før jeg er restitueret /er kommet mig fuldstændigt efter en arbejdsdag	It takes me over an hour before I am fully recovered/fully improved after a work day
Item 8	Når jeg kommer hjem efter arbejde, skal folk lade mig være i et stykke tid	When I get home after work, people have to leave me alone for a while
Item 9	Efter en arbejdsdag er jeg for træt til at begynde andre aktiviteter	After a working day I am too tired to begin other activities
**Response Categories**		**Response Categories**
1. Aldrig 2. Sjældent 3. Engang i mellem 4. For det meste 5. Altid		1. Never 2. Rarely 3. Sometimes 4. Generally 5. Always

**Table 2 ijerph-16-02334-t002:** Items used in our assessment of construct validity. Selection is based on the similarity of construct with that of the NFR Scale.

English Translation	Scoring Values
How many days in the last four weeks has muscle or joint pain inhibited you? (e.g., affected you daily routine or activities)	0–28 days
How physically demanding do you normally perceive your working situation? [27]	0–10 numerical ratings scale. 0 being not demanding; 10 being maximally demanding
How do you rate your overall health? [24]	5 point likert scale 1 being very poor; 5 being excellent
How do you rate your current work ability in relation to the psychological/cognitive demands of your work? [26]	5 point likert scale 1 being very poor; 5 being excellent
How do you rate your current work ability in relation to the physical demands of you work? [26]	5 point likert scale 1 being very poor; 5 being excellent
How do you rate your current work ability? [26]	0–10 numerical ratings scale. 0 being unable to work; 10 being work ability at it’s best
Do you wake up fresh and recovered? [25]	6 point likert scale 1 being at no time; 6 being all the time
Do you feel calm and relaxed? [25]	6 point likert scale 1 being at no time; 6 being all the time

**Table 3 ijerph-16-02334-t003:** Primary descriptive statistics of included participants from the PIPPI (*n* = 344) and PRIO (*n* = 765) projects.

Demographics Information	PIPPI ^a^		PRIO ^b^	
	Mean or *n*	(SD) or (%)	Mean or *n*	(SD) or (%)
Sex (male)	242	71%	74	9.7%
Age (years)	44	SD 10	43	SD 11
BMI (kg/m^2^)	26.6	SD 4.4	25	SD 3.9
Smoking				
Daily	80	23%	N/A	
Never	27	8%	360	48%
Former	97	28%	237	31%
Current	140	41%	160	21%
Self-reported health				
Very good	29	8%	49	6%
Good	122	36%	317	42%
Fairly good	166	48%	329	44%
Poor	26	8%	60	8%
Very poor	1	0.3%	2	0.3%
NFR index	51	SD 8.8	55	SD 12.6

PIPPI = Participatory Intervention on Physical and Psychosocial resources of Industrial workers; PRIO = Prioritized Working Hours [Prioritet Arbejdstid]; ^a^ Participants were involved primarily with work in the Danish manufacturing industry, ^b^ Participants were involved primarily with shift-work in psychiatric and somatic healthcare settings.

**Table 4 ijerph-16-02334-t004:** Summary descriptive statistics of participant responses to the scales from the PIPPI ^a^ (*n* = 344) project used to assess construct validity.

Construct Validity Items	Mean	SD or *n* (%)
Overall work ability	8	SD 1.4
Work ability in the physical domain		
Very poor	72	21%
Poor	157	46%
Fairly good	100	29%
Good	15	4%
Very good	0	0%
Work ability in the psychological domain		
Very poor	56	16%
Poor	155	45%
Fairly good	47	14%
Good	24	7%
Very good	1	0.3%
Feeling recovered on awaking		
At no time	8	2%
Rarely	103	30%
Some of the time	153	45%
Most of the time	60	17%
All of the time	20	6%
Feeling calm and relaxed		
At no time	14	4%
Rarely	144	42%
Some of the time	95	28%
Often	59	17%
Most of the time	23	7 %
All of the time	9	3%
Physical exertion at work	6	SD 2.3
Days of inhibiting pain		
0–10	287	85%
11–20	27	8%
>20	24	7%

^a^ Participants were involved primarily with work in the Danish manufacturing industry.

**Table 5 ijerph-16-02334-t005:** The distribution of missing responses among cases removed by list-wise deletion.

NFR Item	PIPPI *n* = 11	PRIO *n* = 40
Item 1	1	6
Item 2	1	2
Item 3	2	14
Item 4	6	14
Item 5	0	5
Item 6	1	6
Item 7	1	9
Item 8	1	3
Item 9	0	3

*n* = number of cases deleted. For details of each item please refer to Table 1.

**Table 6 ijerph-16-02334-t006:** Construct validity of the Danish 9-item Need for Recovery Scale and several short-form versions against other related items (*n* = 344).

Related Construct	Danish Need for Recovery (NFR) Scale						
	Primary Aim					Secondary Aim	
	Items 1–9 (Original)	Items 1, 2, 5–9	Items 2, 6, 7	Items 2, 6	Item 7	Items 1, 2 & 9	Items 2, 9
Days of inhibiting pain ^k^	0.28 (0.20, 0.35)	0.30 (0.23, 0.37)	0.31 (0.24, 0.38)	0.31 (0.24, 0.38)	0.27 (0.18, 0.35)	0.32 (0.24, 0.39)	0.33 (0.24, 0.40)
Perceived exertion ^p^	0.17 (0.06, 0.28)	0.18 (0.08, 0.29)	0.21 (0.10, 0.31)	0.22 (0.11, 0.31)	0.17 (0.07, 0.27)	0.26 (0.14, 0.36)	0.26 (0.15, 0.35)
Perceived health ^p^	−0.32 (−0.41, −0.21)	−0.36 (−0.44, −0.26)	−0.32 (−0.41, −0.22)	−0.31 (−0.41, −0.21)	−0.28 (−0.37, −0.19)	−0.35 (−0.44, −0.26)	−0.36 (−0.45, −0.26)
Mental work ability ^k^	−0.30 (−0.38, −0.22)	−0.32 (−0.40, −0.24)	−0.28 (− 0.36, −0.20)	−0.29 (−0.37, −0.21)	0.26 (0.17, 0.34)	−0.31 (−0.39, −0.23)	−0.26 (−0.34, −0.18)
Physical work ability ^k^	−0.22 (−0.30, −0.14)	−0.27 (−0.34, −0.19)	−0.25 (−0.33, −0.17)	−0.25 (−0.32, −0.16)	−0.23 (0.14, 0.32)	−0.28 (−0.35, −0.19)	−0.28 (−0.35, −0.19)
Overall work ability ^k^	−0.23 (−0.30, −0.15)	−0.27 (−0.34, −0.19)	−0.24 (−0.31, −0.16)	−0.23 (−0.30, −0.15)	−0.23 (0.14, 0.31)	−0.27 (−0.35, −0.19)	−0.27 (−0.34, −0.18)
Feeling rested on waking ^p^	−0.48 (−0.56, −0.38)	−0.48 (−0.57, −0.39)	−0.44 (−0.53, −0.34)	−0.39 (−0.49, −0.27)	−0.45 (−0.54, −0.35)	−0.43 (−0.52, −0.34)	−0.40 (−0.50, −0.29)
Feeling calm and relaxed ^k^	−0.40 (−0.47, −0.32)	−0.43 (−0.50, −0.36)	−0.40 (−0.47, −0.32)	−0.40 (−0.47, −0.32)	−0.38 (−0.46, −0.29)	−0.40 (−0.47, −0.31)	−0.37 (−0.45, −0.29)

Primary aim: reduction of scale utilising any NFR item; Secondary aim: reduction of scale using only items 1, 2 and 9. For details of each item and outcome please refer to Table 1 and Table 2 respectively. Values: correlation (95% CI); ^k^ denotes Kendall’s tau; ^p^ denotes Pearson’s r.

**Table 7 ijerph-16-02334-t007:** Criterion Validity and Responsiveness of several short-form versions of the Danish Need for Recovery Scale against the full 9-item version—exploratory analyses.

Aim being Addressed	NFR Scale Items Used	Criterion Validity ICC (95% CI) (*n* = 344)	Responsiveness ICC (95% CI) (*n* = 245)
Primary Aim	Items 1, 2 & 5–9	0.92 (0.90, 0.93)	0.94 (0.92, 0.95)
	Items 2, 6 & 7	0.85 (0.82, 0.88)	0.79 (0.74, 0.84)
	Items 2 & 6	0.82 (0.79, 0.85)	0.74 (0.68, 0.79)
	Item 7	0.67 (0.60, 0.72)	0.51 (0.41, 0.60)
Secondary Aim	Items 1, 2 & 9	0.83 (0.80, 0.86)	0.76 (0.70, 0.81)
	Items 2 & 9	0.66 (0.59, 0.71)	0.67 (0.60, 0.73)

Primary aim: reduction of scale utilising any NFR item; Secondary aim: reduction of scale using only items 1, 2 and 9. ICC = Intra-class correlation coefficient. For details of each item please refer to Table 1.

**Table 8 ijerph-16-02334-t008:** Criterion Validity and Responsiveness of several short-form versions of the Danish Need for Recovery Scale against the full 9-item version—confirmatory analyses.

Aim being Addressed	NFR Scale Items Used	Criterion Validity ICC (95% CI) (*n* = 765)	Responsiveness ICC (95% CI) (*n* = 475)
Primary Aim	Items 2, 6 & 7	0.88 (0.86, 0.90)	0.80 (0.76, 0.83)
	Items 2 & 6	0.82 (0.80, 0.84)	0.72 (0.67, 0.76)
Secondary Aim	Items 1, 2 & 9	0.86 (0.84, 0.88)	0.73 (0.69, 0.77)

Primary aim: reduction of scale utilizing any NFR item; Secondary aim: reduction of scale using only items 1, 2 and 9. ICC = Intra-class correlation coefficient. For details of each item please refer to Table 1.

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
