# Peer review of "Validation of a Short-Form Version of the Danish Need for Recovery Scale against the Full Scale"

_ijerph, 2019, doi:10.3390/ijerph16132334_

Reviewer 1 Report

Well done, the manuscript conveys rigor and utility of your work. Future researchers will be pleased that a short version of the Need for Recover Scale is available and valid. The manuscript is very well written and reads well. I have no recommendations for changes or edits. 

Author Response

Well done, the manuscript conveys rigor and utility of your work. Future researchers will be pleased that a short version of the Need for Recover Scale is available and valid. The manuscript is very well written and reads well. I have no recommendations for changes or edits. 

1.      We thank you for your review.

Reviewer 2 Report

The paper is mostly well-written and presented, and deals with an interesting topic. Conclusions section is very short. Findings may not be generalised to other countries.

Author Response

The paper is mostly well-written and presented, and deals with an interesting topic. Conclusions section is very short. Findings may not be generalised to other countries.

1.      We thank you for your review. We agree with your point and have added the sentence “Any generalization of these findings to other countries ought to be made with caution” to the conclusion of the manuscript.

Reviewer 3 Report

A paper validating a short-form Danish version of the Need for Recovery Scale, as well as further testing of clinimetric properties of this version of the tool could be useful.  However, I feel that due to some issues with detail, and a misrepresentation of what was done in this study means that it is not publishable in its current form.

Specifically, I feel the title does not seem to represent the contents of the study.  Also, the introduction is poor, needs to present a better rationale for the study and the methodology employed. 

Overall the design was poor. Research question is not well designed and methods carried out with insufficient detail and information to replicate. Also, there does not appear to be any justification for the method used for calculation of simple size.  The sample is very small to do the statistical analysis. The criteria of inclusion and exclusion were not included in the text. You did not include a flow diagram of the process. The statistics being used, and the underlying purpose for the various analyses is confusing and needs to be made clear.  For example, you purport to validate, evaluate test-retest, look at construct validity amongst other things, but the statistics for each is confusing.

Results, Discussion and Conclusions sections are not informative. I dont believe this study adds a great deal of novel and new information.

I regret that the disposition is not favorable, but would like to thank you for your support.

To further strengthen the article, I would also recommend to:

·         Provide the rationale of the need for a validating a short-form Danish version of the Need for Recovery Scale

·         The validity of the tool is reliant on what it is purported to measure, therefore I think insufficient detail and rationale for the validating a short-form Danish version of the Need for Recovery Scale has been provided.  There is no detail around how this tool differs from Need for Recovery Scale measures, or if it is better?  What rationale is there for using this measure?  For which particular circumstances has the Need for Recovery Scale been devised? (ie research versus clinical application).

Author Response

A paper validating a short-form Danish version of the Need for Recovery Scale, as well as further testing of clinimetric properties of this version of the tool could be useful.  However, I feel that due to some issues with detail, and a misrepresentation of what was done in this study means that it is not publishable in its current form.

1.      We thank you for your review. We have done our best to rectify the manuscript in accordance to your specific points below, which we think have improved the quality of the paper.

Specifically, I feel the title does not seem to represent the contents of the study. 

2.      In this study we validated a shortened version of the Danish Need for Recovery (NFR) Scale as an adequate representation of the full Danish NFR Scale. To make the title more representative we have modified it to “Validation of a short-form version of the Danish Need for Recovery Scale against the full Scale”.

Also, the introduction is poor, needs to present a better rationale for the study and the methodology employed. 

3.      The primary rationale statement for conducting this study is that “Doing so would greatly decrease the burden of the NFR Scale in research and thus increase its feasibility of use in future cohorts and studies.” (page 2, line 29). We have modified the introduction to strengthen the argumentation for the rationale of the study.

4.      To keep the introduction short and clear, we did not include methodological aspects there, but presented them in the methods section. However, upon reviewing your comments we agree that it is unclear whether the aim is to validate this shortened version independently of the full scale (i.e. treat it as if it is a brand new scale being developed) or validate it as an adequate representation of the full scale (i.e. what we have done using methods for validating item-reduction of scales). As such the line “an adequate representation of the full scale” has been added to the aim statements.

         Therefore, the aim of this study is to create and validate a shortened version of the Danish NFR Scale that is an adequate representation of the full scale. A secondary aim is to validate a specific shortened Danish version of the NFR Scale that has been used in previous studies [21,22]as an adequate representation of the full scale.

Overall the design was poor.

5.      We do not follow how you mean the design of the study was poor as the design of the study follows established recommendations for scale‑reduction [1][2] (information provided on page 4, line 2). We interpret your comment that it may be caused by us not being sufficiently clear on whether we are conducting an independent validation of the short-form versions or validating the short-form versions as an adequate representation of the full scale as discussed in point 4. We hope that our changes in making this point clearer in the manuscript have clarified this issue.

Research question is not well designed and methods carried out with insufficient detail and information to replicate.

6.      We have tried to balance the information given, so that it gives sufficient information to permit replication without giving too many details that will make the manuscript hard to follow. With the comment in mind, we have now made added more detail to the statistical analyses sections of the manuscript to ensure repeatability.

Also, there does not appear to be any justification for the method used for calculation of simple size.  The sample is very small to do the statistical analysis.

7.      You are right that the sample size calculation was not performed a priori, but that the analyses were based upon data available. To meet your comment, we have made a post-hoc sample size calculation using our analysis of criterion validity (the primary measure of whether the short-form scales adequately represent the full scale). In this case, the minimum sample size required to achieve 80% power, given an agreement of 0.75 and an alpha of 0.05 is 75 [3]. With the number of participants included in our analyses (1109 counting both exploratory and confirmatory analyses) we believe our sample size is more than adequate for the analyses conducted. This information has been added to the manuscript.

The criteria of inclusion and exclusion were not included in the text.

8.      The criteria for inclusion for each of the primary studies (PIPPI and PRIO) included in this study provided on page 3, line 2 and page 3 line 7 respectively. We have added an additional statement that “all participants from PIPPI and PRIO that provided any information regarding NFR at baseline were included in this study.”

You did not include a flow diagram of the process.

9.      A flow chart depicting the analyses and decision processes has been added.

The statistics being used, and the underlying purpose for the various analyses is confusing and needs to be made clear.  For example, you purport to validate, evaluate test-retest, look at construct validity amongst other things, but the statistics for each is confusing.

10.    To clarify the statistics used and the purposes for them modifications have been made to the statistical analyses sections (sections 2.3-2.5; page 4) of the manuscript.

Results, Discussion and Conclusions sections are not informative. I dont believe this study adds a great deal of novel and new information.

11.  This study shows that a shortened version of the Danish NFR scale, consisting of the 3 items described in the paper, maintains very similar psychometric properties to the full scale. As reducing the number of items in the NFR scale has not been previously conducted, we believe that this study adds new and very relevant information to the literature – particularly for future studies considering or planning to use the NFR.

 I regret that the disposition is not favorable, but would like to thank you for your support.

To further strengthen the article, I would also recommend to:

·         Provide the rationale of the need for a validating a short-form Danish version of the Need for Recovery Scale

12.  Thank you, please see our response number 3

·         The validity of the tool is reliant on what it is purported to measure, therefore I think insufficient detail and rationale for the validating a short-form Danish version of the Need for Recovery Scale has been provided. 

There is no detail around how this tool differs from Need for Recovery Scale measures, or if it is better? 

13.  We believe there is a lot of detail in this study on how similar the shortened versions are to the full NFR scale (results for construct and criterion validity and responsiveness) – which is the purpose and main novelty of this paper.

What rationale is there for using this measure? 

14.  The rationale behind using a shortened version of the questionnaire is that you are able to get the same information as the full scale, while only using 3 items instead of 9. This will reduce the burden to participants, and thus increase the feasibility for future surveys, cohorts and intervention studies to include the NFR (which has often not been included due to the many items and participant burden). This point has been made clearer in the introduction.

 For which particular circumstances has the Need for Recovery Scale been devised? (ie research versus clinical application).

15.  The NFR scale was originally designed for research purposes and although some clinical settings have used it (work surveys etc) the NFR scale is still primarily a research tool.

References

1.        Goetz, C.; Coste, J.; Lemetayer, F.; Rat, A.C.; Montel, S.; Recchia, S.; Debouverie, M.; Pouchot, J.; Spitz, E.; Guillemin, F. Item reduction based on rigorous methodological guidelines is necessary to maintain validity when shortening composite measurement scales. J. Clin. Epidemiol. 2013, 66, 710–718.

2.        de Vet, H.C.W.; Terwee, C.B.; Mokkink, L.B.; Knol, D.L. Measurement in Medicine; Cambridge University Press: Cambridge, 2011; ISBN 9780511996214.

3.        Temel, G.; Erdogan, S. Determining the sample size in agreement studies. Marmara Med. J. 2017, 30, 101–112.

Round  2

Reviewer 3 Report

I think the authors addressed the concerns. It is ready for publication.